REVIEW-SYMPOSIUM

# Incretin hormones and obesity

Constanza Alcaino ⬤, Frank Reimann and Fiona M. Gribble ⬤

*Institute of Metabolic Science Metabolic Research Laboratories, University of Cambridge, Addenbrooke's Hospital, Cambridge, UK*

Handling Editors: Kim Barrett & Stephen Keely

The peer review history is available in the Supporting Information section of this article (https://doi.org/10.1113/JP286293#support-information-section).

**Abstract figure legend** The incretin system in obesity. The incretin hormones glucose-dependent insulinotropic polypeptide (GIP) (yellow) and glucagon-like peptide-1 (GLP-1) (blue) are produced by the proximal and distal small intestinal epithelium, where they are released postprandially into the bloodstream to modulate a myriad of physiological and metabolic functions. GIP is mostly produced by K-cells in the duodenum and jejunum, whereas GLP-1 is produced by L-cells mostly in the ileum and large intestine. These incretin hormones act on the pancreas causing an increase in insulin secretion. GIP increases glucagon production, whereas GLP-1 decreases it. These hormones have been shown to reduce food intake, causing weight loss. Central GLP-1 increases nausea, whereas GIP has been shown to decrease nausea by acting on GIP receptors in the brainstem.

**Constanza Alcaino** is a Research Associate at the Institute of Metabolic Science (IMS) of the University of Cambridge, working under the mentorship of Professors Fiona Gribble and Frank Reimann. During her previous postdoctoral training, she worked at the Enteric Neuroscience Program (ENSP) of Mayo Clinic, investigating the role of the ion channel Piezo2 in enterochromaffin cell mechanosensitivity. Constanza's current work focuses on understanding the molecular pathways involved in chemical and mechanical sensing by enteroendocrine cells that lead to hormone release and modulate physiology.

**Abstract** The incretin hormones glucagon-like peptide-1 (GLP-1) and glucose-dependent insulinotropic polypeptide (GIP) play critical roles in co-ordinating postprandial metabolism, including modulation of insulin secretion and food intake. They are secreted from enteroendocrine cells in the intestinal epithelium following food ingestion, and act at multiple target sites including pancreatic islets and the brain. With the recent development of agonists targeting GLP-1 and GIP receptors for the treatment of type 2 diabetes and obesity, and the ongoing development of new incretin-based drugs with improved efficacy, there is great interest in understanding the physiology and pharmacology of these hormones.

(Received 28 August 2024; accepted after revision 31 October 2024; first published online 22 November 2024)

**Corresponding authors** F. Reimann and F. M. Gribble: Institute of Metabolic Science Metabolic Research Laboratories, University of Cambridge, Addenbrooke's Hospital, Cambridge CB2 0QQ, UK.      Email: fr222@cam.ac.uk and fmg23@cam.ac.uk

## Introduction

The gut epithelium lines the length of the gastro-intestinal (GI) tract, forming a barrier between the internal and external environments. It serves important roles in food digestion and absorption, and generates signals related to nutrient ingestion and microbial metabolism via the production of gut hormones. These hormones are produced by a specialised type of epithelial cell known as the enteroendocrine cell (EEC). Nutrients and other chemicals found in the intestinal lumen classified as odorants, irritants, bacterial metabolites, secondary bile acids and mechanical forces can stimulate EEC activation and hormone release (Alcaino et al., 2018; Bellono et al., 2017; Goldspink et al., 2018, 2020; Larraufie et al., 2018; Miedzybrodzka et al., 2021; Parker et al., 2009).

Gut hormones control a myriad of metabolic and physiological functions, and send local paracrine signals to neighbouring cells and neurons to control GI function (Bany Bakar et al., 2023). Within the GI tract, EEC hormones regulate intestinal motility, secretion, sensation, nutrient absorption, gastric emptying and epithelial growth (Bayrer et al., 2023; Gorboulev et al., 2012; Hargrove et al., 2020; Nozawa et al., 2009; Treichel et al., 2022; Wang et al., 2017), as well as pancreatic enzyme secretion and gallbladder contraction (Mawe, 1998; Takano & Yule, 2023). Their ability to activate extrinsic afferent nerves suggests the intestinal epithelium might be a key player in the treatment of disorders of the gut–brain axis (Kaelberer et al., 2018; Lu et al., 2019). Outside of the GI tract, several gut hormones have been shown to be critical for the control of metabolism and modulation of food intake (Lewis et al., 2020, 2022; Martin et al., 2017b; Mawe & Hoffman, 2013). The incretin hormones glucagon-like peptide-1 (GLP-1) and glucose-dependent insulinotropic polypeptide (GIP), for example, increase postprandial insulin secretion (Nauck & Müller, 2023), with GLP-1 being shown to lower plasma glucose in people with type 2 diabetes (Nauck

et al., 1993), which led to the development and surge in clinical use of GLP-1 receptor (GLP1R) agonists and, more recently, GLP1R/GIP receptor (GIPR) dual agonists for the treatment of type 2 diabetes and obesity (Gallwitz, 2022; Guccio et al., 2022; Nauck & Müller, 2023).

In this review, we aim to describe the roles of gut-derived incretin hormones in metabolism and the treatment of obesity.

**Intestinal epithelial EECs.** Intestinal EECs make up ∼1% of the intestinal epithelium, but, in total, they constitute the largest site of hormone production in the human body (Beumer, Puschhof, et al., 2020). EEC subtypes have been classified according to the hormone(s) they produce, as determined by immunostaining. L-cells producing GLP-1 are most common in the ileum where they co-produce peptide YY (PYY), and in the colon/rectum where they co-produce PYY, neurotensin and insulin-like 5 (INSL5), with lower L-cell numbers in the proximal small intestine (Billing et al., 2019; Roberts et al., 2019). K-cells producing GIP are mainly located in the upper small intestine and form a relatively distinct cell cluster, as evident from single cell RNA sequencing (Bai et al., 2022; Beumer, Puschhof, et al., 2020; Hayashi et al., 2023; Smith et al., 2024) Enterochromaffin (EC) cells are the most abundant type of EEC and regulate intestinal motility, secretion and sensation by producing serotonin (5-HT); EC-cells also express the tachykinin gene, *Tac1*, which can be processed to substance P, well known for its role in nociception, but the motility defects seen after EC-cell ablation could be at least partly restored with exogenous serotonin(-precursor) administration (Wei et al., 2021) and 5-HT$_3$-receptor antagonist blocked the propulsive effects of INSL5, a distal L-cell hormone that targets EC-cells in the mouse colon (Koo et al., 2022). Other described EEC subtypes include D-cells (somatostatin, SST), I-cells (cholecystokinin, CCK), M/X-cells (motilin, MLN; ghrelin, GHRL), N-cells

(NTS) and S-cells (secretin, SCT) (Bany Bakar et al., 2023). Because many EECs switch on SCT production when maturing along the crypt villus axis (Beumer et al., 2018), the existence of a specific S-cell has been questioned (Hysenaj et al., 2024), although a distinct S-cell population was identified in humans, closely related to EC-cells (Hickey et al., 2023). Transcriptomic analyses have revealed substantial overlaps between some EEC populations, particularly those producing GLP-1, PYY, CCK, NTS, SCT and INSL5 (depending on location), making the traditional alphabetical EEC nomenclature somewhat imperfect, and raising challenges in any attempt to target secretion of a specific gut hormone (Habib et al., 2012). These transcriptomic results have been backed up by liquid chromatography-tandem mass spectrometry analysis of purified EECs from native tissue or intestinal organoids, which similarly identified overlap in the production of different peptide hormones. Purified murine *Cck*-expressing cells, for example, contained detectable SCT, GIP, GLP-1, PYY and NTS peptides (Egerod et al., 2012) and purified human *GCG*-expressing cells (i.e. labelled for GLP-1 biosynthesis) from ileal organoids, were enriched for production of PYY, NTS, pancreatic polypeptide and urocortin 3 (Goldspink et al., 2020).

In the distal colon, GLP-1, PYY and INSL5 were found to be co-located in the same vesicular pool, when analysed by super-resolution microscopy in murine and human primary cultures, and were co-released in response to a range of stimuli (Billing et al., 2018). Other studies have similarly reported the co-location of several hormones within the same EECs (GLP-1, SCT, CCK, NTS, GHRL and 5-HT) in different combinations and sometimes in separate subcellular vesicular pools (Fothergill et al., 2017; Grunddal et al., 2016; Nilsson et al., 1991). However, there is currently no strong evidence that individual EECs differentially traffic peptide hormones into distinct vesicular pools, and incidences of hormonal staining appearing in different vesicles may reflect either vesicular pools formed at different stages in a cell's lifetime, or artefacts of immunostaining and image analysis.

Glucagon-like peptide-2 (GLP-2) is co-released by L-cells as it is generated alongside GLP-1 by prohormone convertase cleavage of the precursor proglucagon peptide. Its best understood role is to maintain and repair the epithelial barrier (Benjamin et al., 2000). Clinically, GLP-2 analogues are used for the treatment of short bowel syndrome (Burness & McCormack, 2013). A few studies have reported that GLP-2 modulates glucagon secretion and glucose homeostasis, but its importance for the regulation of metabolism and food intake is inconclusive (Bahrami et al., 2010; Lund et al., 2011; Sorensen et al., 2003).

Gradients of gut hormone production along the GI tract are well established (Martin et al., 2017a; Roberts et al., 2019) and are considered to arise from the properties of the stem cells from which the EECs are generated because stem cells from different gut regions when differentiated into organoids *in vitro* generate EECs typical of the intestinal region from which they originated (Beumer, Gehart, et al., 2020). They also undergo maturation and switches in hormone production during EEC maturation, accompanying the movement of cells out of the crypt domain and into villi (Beumer et al., 2018; Roth & Gordon, 1990). Gradients of the transcription factor BMP4 along the crypt–villus axis appear to be one local factor contributing to switching the repertoire of hormones expressed by maturing EECs (Beumer et al., 2018). The use of intestinal organoid technology has greatly enhanced our understanding of EEC development and function because they closely recapitulate native epithelial physiology; they can also be genetically modified to enable fluorescent EEC labelling, manipulation and identification in long-term culture (Beumer, Gehart, et al., 2020; Beumer et al., 2022; Guccio et al., 2025; Miedzybrodzka et al., 2020; Petersen et al., 2014).

**Nutrient sensing by incretin-producing EECs.** K- and L-cells play important roles as nutrient sensors in the intestinal epithelium and the molecular mechanisms linking nutrient sensing to hormone release have been extensively studied in cell lines, intestinal organoids, primary cultures and perfused intestinal models (Santos-Hernandez et al., 2024).

GIP is highly abundant in duodenal K-cells, where its secretion is stimulated by all major macronutrients following their digestion and absorption in the proximal intestine (Guccio et al., 2022). Elevated plasma GLP-1 levels are also detectable within 5 min after oral glucose ingestion, and usually reach a peak 30–60 min after food ingestion, depending on the meal composition (Herrmann et al., 1995). This rapid initial rise in GLP-1 has been attributed to the small population of L-cells located in the duodenum and jejunum (Panaro et al., 2020; Song et al., 2019), even though more L-cells are found in the ileum and colon (Beumer, Puschhof, et al., 2020). Distally-located L-cells do not normally make contact with rapidly-absorbable ingested nutrients, but L-cells in the ileum probably account for the sustained postprandial release of GLP-1 following ingestion of slowly digestible foods, and, along with the populations of colonic and rectal EECs, are assumed to be activated by bacterial metabolites, bile acids and lipopolysaccharides (Brighton et al., 2015; Chimerel et al., 2014; Elliott et al., 1993; Goldspink et al., 2018; Lebrun et al., 2017; Panaro et al., 2020; Thomas et al., 2009; Tolhurst et al., 2012). Activation of L-cells in the distal small intestine also underlies the dramatic elevations in postprandial GLP-1 and PYY concentrations observed after gastric bypass surgery (Jørgensen et al., 2012; Larraufie et al., 2019).

EEC detection of stimuli depends on G-protein coupled receptors (GPCRs) and nutrient transporters located on the apical and/or basolateral sides of EECs (Fig. 1), which initiate intracellular signalling pathways that usually involve $Ca^{2+}$ and/or cAMP (Santos-Hernandez et al., 2024). They express a repertoire of GPCRs responsive to a wide range of macronutrient digestion products including long chain fatty acids, 2-monoacylglycerols and amino acids, as well as to short-chain fatty acids, bile acids, neurotransmitters, hormones and mechanical forces (Alcaino et al., 2018; Bellono et al., 2017; Christiansen et al., 2019; Guccio et al., 2025; Jepsen et al., 2019; Lu et al., 2018, 2019).

GPCRs leading to hormone release are usually G$\alpha$s- or G$\alpha$q-coupled, activation of which leads to elevation of cAMP or $Ca^{2+}$, respectively. In EECs, G$\alpha$s-coupled receptors such as the monoacylglycerol receptor GPR119 and the bile acid receptor GPBAR1 have been shown to increase intracellular cAMP and induce hormone release, playing important physiological roles after fat ingestion (Brighton et al., 2015; Hodge et al., 2016). The free fatty acid receptor FFAR1 is activated by long chain fatty acids following triglyceride ingestion and is a key G$\alpha$q-coupled EEC receptor linked to increases in intracellular $Ca^{2+}$ and release of GLP-1, CCK and GIP (Goldspink et al., 2018; Gribble et al., 2017; Guccio et al., 2025; Shah et al., 2012). Other important G$\alpha$q-coupled receptors in K- and L-cells are the short-chain fatty acid receptor FFAR2, responsive to acetate, butyrate and propionate, and CASR and GPR142, responsive to aromatic amino acids such as tryptophan and phenylalanine (Goldspink et al., 2018, 2020; Guccio et al., 2025; Rudenko et al., 2019).

Electrogenic brush border substrate transporters are another family of membrane proteins critical for EEC nutrient sensing. The sodium-glucose transporter 1 (SGLT1) acts as the intestinal sensor of ingested glucose

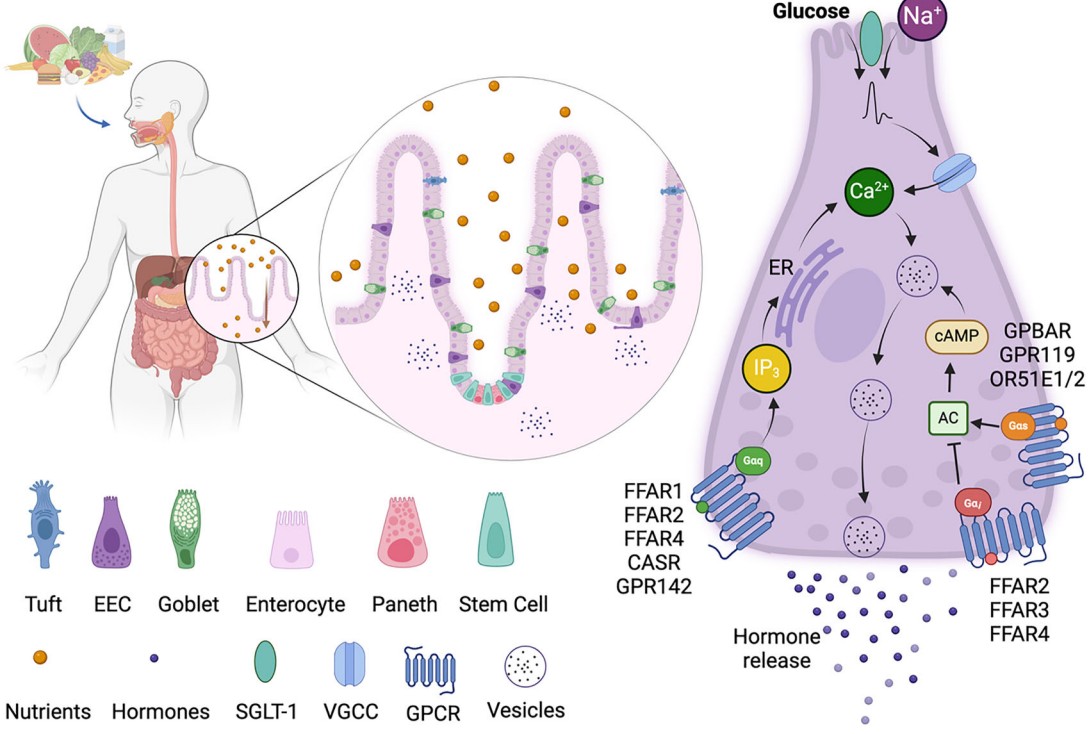

**Figure 1. Nutrient sensing by enteroendocrine cells**
Carbohydrates, amino acids, fats and bile acids can activate small intestinal EECs either from the apical (top) or basolateral (bottom) sides. Glucose is transported alongside $Na^+$ ions by SGLT1, which can cause a membrane depolarisation that activates VGCCs, bringing $Ca^{2+}$ inside the cell and evoking the release of hormones via vesicular exocytosis. Fatty acids and amino acids can activate GPCRs of the G$\alpha$q type (FFAR1, FFAR2, FFAR4 for long- and short-chain fatty acids and CASR and GPR142 for aromatic amino acids). G$\alpha$q coupling induces an increase of IP3, which activates IP3R in the ER, also inducing $Ca^{2+}$ increase and hormone release. Some fatty acid receptors are also G$\alpha$i-coupled (FFAR2, FFAR3 and FFAR4), which inhibit adenyl-cyclase (AC) and cAMP production. By contrast, some medium-chain fatty acid (OR51E1/E2), bile acid (GPBAR1) and monoacylglycerol (GPR119) receptors increase cAMP via G$\alpha$s-coupling and induce hormone release. Abbreviations: SGLT1, sodium-coupled glucose cotransporter 1; VGCCs, voltage-gated calcium channels; ER, endoplasmic reticulum; cAMP, cyclic adenosine monophosphate; AC, adenylyl cyclase; GPCRs, G-protein coupled receptors; FFAR1–4, free fatty acid receptor 1–4; CASR, calcium sensing receptor; IP3, inositol phosphate 3; OR51E1/E2, olfactory receptor 51E1 and E2; GPBAR1, G-protein coupled bile acid receptor 1.

in K- and L-cells because it carries glucose and $Na^+$ ions across the apical membrane, generating membrane depolarisation that activates voltage-gated $Ca^{2+}$ channels and $Ca^{2+}$ influx, leading to GLP-1 release (Gorboulev et al., 2012). Facilitative glucose transporters of the GLUT family, also expressed in L- and K-cells, equilibrate glucose concentrations across the basolateral membrane and dominate the regulation of intracellular glucose concentrations in EECs, but do not appear to mediate glucose-triggered incretin hormone secretion (Parker et al., 2012). The same transporters are expressed by neighbouring enterocytes to enable the transepithelial absorption of glucose.

Paracrine communication is an additional regulator of hormone release from intestinal EECs, in both the fasting state and after a meal. For example, the release of SST by D-cells plays a dynamic role in the local inhibition of GLP-1 (and probably other gut hormones) secretion (Jepsen et al., 2019), whereas GLP-1 secretion from L-cells has been shown to induce 5-HT release from EC-cells (Lund et al., 2018).

**Local gastrointestinal effects of incretin hormones.** EEC-derived hormones, such as MLN and 5-HT are known to be important regulators of intestinal motility in fasted (Foreman et al., 2024; Mori et al., 2023) and fed states (Mawe & Hoffman, 2013), whereas CCK has been shown to inhibit gastric emptying and to modulate gastric acid secretion (Fried et al., 1991; Kanagawa et al., 2002). Similarly, exogenous administration of GIP slowed small intestinal transit by ∼40%, as well as glucose absorption, through an SST-dependent mechanism (Ogawa et al., 2011).

GLP-1 derived from L-cells plays an important role in the regulation of gastric emptying (Song et al., 2019) and, alongside PYY, is responsible for the 'ileal brake', a mechanism that dynamically controls the rate of nutrient delivery from the stomach into the proximal small intestine (Wettergren et al., 1993). The ileal brake is considered to be mediated by vagal innervation because selective knockdown of *Glp1r* in vagal afferent neurons not only increased food intake, but also accelerated gastric emptying in a rat model (Krieger et al., 2016). In the large intestine, GLP-1 and PYY are co-released with INSL5, which has been shown to be an important regulator of colonic motility, through a mechanism that probably involves 5-HT release and 5-HT$_3$ receptors from/in EC cells or enteric neurons (Billing et al., 2019; Koo et al., 2022; Lewis et al., 2020).

**Incretin hormones and glucose homeostasis.** Incretin hormones play a crucial role in the control of glucose homeostasis. They are released after nutrient ingestion and account for over 60% of physiological postprandial insulin secretion (Dupre et al., 1973; Vilsbøll et al., 2003). The greater observed insulin release after oral or intestinal glucose delivery compared with i.v. infusion, is known as the incretin effect, and lines up with the findings that only oral but not i.v. glucose administration causes the release of the incretin hormones GLP-1 and GIP (Cataland et al., 1974). Studies in healthy individuals show that after an oral glucose tolerance test or mixed meal, inhibiting GIPR with the antagonist GIP(3-30)NH$_2$ has a more robust inhibitory effect on postprandial insulin secretion than inhibiting GLP1R with exendin-9 (Ex9) (Gasbjerg et al., 2020), suggesting a particularly important physiological role for endogenous GIP in underlying the incretin effect in healthy humans.

GLP-1 and GIP act on G$\alpha$s-coupled GLP1R and GIPR on pancreatic $\beta$ cells, causing cAMP elevation and potentiation of $Ca^{2+}$-dependent insulin release (Skelin & Rupnik, 2011). Unlike sulfonylureas, which increase insulin release largely independently of circulating glucose levels, physiological concentrations of incretin hormones do not stimulate insulin secretion at low glucose plasma levels, therefore posing a minimal threat of inducing hypoglycaemia when used in a clinical setting (Siegel et al., 1992). Both incretins lose efficacy in the context of chronic elevated blood glucose seen in diabetes, but the somewhat better-preserved effectiveness of GLP-1 compared to GIP has been linked to GIPR-downregulation (Zhou et al., 2007) or alternatively to a Gs to Gq-coupling switch downstream of the GLP-1, but not the GIP-receptor (Oduori et al., 2020).

Coadministration of glucose along with GIP and GLP-1 to mimic postprandial plasma concentrations induced an increase in insulin in both cases, but only GLP-1 suppressed glucagon (de Heer et al., 2008; Vilsbøll et al., 2003). This inhibitory effect of GLP-1 on glucagon is considered to be partially the result of a direct effect on pancreatic $\alpha$ cells and also an indirect pathway involving SST release from pancreatic $\delta$ cells (de Heer et al., 2008). GLP-1 has been linked to longer-term effects on $\beta$ cell mass, insulin synthesis and increased secretion from the exocrine pancreas (Drucker et al., 1987; Hou et al., 2016; Li et al., 2005).

Oxyntomodulin is co-released by L-cells alongside GLP-1 and may contribute to physiological glucose homeostasis and appetite control (Shankar et al., 2018) because it exerts agonistic activity on both GLP-1 and glucagon receptors (Pocai, 2014; Wynne et al., 2006). The metabolic importance of physiological oxyntomodulin release is difficult to disentangle from the effects of GLP-1 and glucagon, but drugs that mimic oxyntomodulin action by acting as dual GLP1R/GCGR agonists are proving highly effective in clinical trials for type 2 diabetes, metabolism dysfunction-associated steatotic liver disease and obesity (Winther & Holst, 2024).

**Effects of gut-derived incretin hormones on food intake and body weight.**  Drugs based on GLP1R agonism, dual GLP1R/GIPR agonism, dual GLP1R/GCGR agonism and triple GLP1R/GIPR/GCGR agonism suppress appetite and reduce body weight in humans, as discussed in the next section. However, the importance of gut-derived GLP-1 and GIP for the physiological control of food intake is less clear.

GLP-1 receptors are located on afferent vagal neurones, as well as in sites in the CNS related to the control of food intake. The latter includes areas that are potentially accessible to circulating hormones, such as circumventricular organs with a compromised blood–brain barrier like the area postrema (AP) and median eminence (ME). Adjacent areas, such as the arcuate nucleus of the hypothalamus (ARH) and the nucleus of the solitary tract (NTS) might also be reached, whereas sites further away such as the paraventricular hypothalamus (PVH) (McLean et al., 2021), the parabrachial nucleus, the amygdala and the lateral septum, which are all labelled in GLP1R-Cre mice (Richards et al., 2014) appear less accessible from the periphery. A study combining inhibition of GLP1R with Ex9, administered either peripherally or centrally, with peripheral or central GLP-1 administration showed that centrally-administered Ex9 was unable to prevent the inhibition of food intake by peripheral GLP-1, whereas peripheral Ex9 did not prevent satiating effects of central GLP-1 (Williams et al., 2009). GLP1R in some brain nuclei might physiologically instead be more receptive to GLP-1 produced in proglucagon expressing neurons found in the brain stem in the NTS and the intermediate reticular nucleus, which project widely throughout the CNS (Holt et al., 2019; Larsen et al., 1997; Merchenthaler et al., 1999). GIPR is not considered to be expressed in the afferent vagus, but has been identified in a number of CNS nuclei, including the hippocampus, PVH, ARH, dorsomedial nuclei of the hypothalamus and NTS (Adriaenssens et al., 2019, 2023; Borner et al., 2021; Costa et al., 2022). Access of peripherally administered fluorescently tagged GIPR-agonists appears to be restricted to circumventricular organs, including the ME and the AP (Adriaenssens et al., 2023), but, unlike GLP-1, there is no known central source of GIP, and the source of ligand for deeply located GIPR in the CNS remains unknown, with GIP-Cre mice failing to label any potential GIP-producing CNS nuclei (Lewis et al., 2024).

Although GLP1R and GIPR in the AP, ME and ARH are readily accessible to peripherally administered incretin receptor agonists because of the leaky blood–brain barrier in these areas (Secher et al., 2014), it is not clear whether concentrations of endogenous incretin hormones are sufficient to target the CNS directly. Active GIP and GLP-1 have very short half-lives in the circulation because of their rapid inactivation by dipeptidyl peptidase 4 (Hansen et al., 1999; Kieffer et al., 1995), suggesting that it does not seem physiologically sensible to release a hormone from the gut only to inactivate it before it reaches its target receptors in the brain. GLP1R located on nerve endings of the afferent vagus in the GI tract, by contrast, are more ideally placed to sense GLP-1 released from the gut, and are considered to be activated by gut-derived GLP-1. For example, some *Glp1r*-expressing nodose ganglia neurones that innervate the stomach and intestine and are sensitive to stretch, and communicate satiety signals to the CNS, potentially providing an opportunity for gut-derived GLP-1 to sensitize these vagal neurons to distension (Williams et al., 2016). A recent study, however, emphasises the importance of *Glp1r*-expression in the hindbrain for pharmacological GLP1R agonist action, with *Glp1r*-expressing neurons in the AP mediating some of the nauseating side effects, whereas selective activation of *Glp1r*-expressing neurons in the NTS induced satiety without aversion in mice (Huang et al., 2024).

Despite these theoretical concerns about the importance of EECs for physiological appetite regulation, gut-restricted stimulation of EECs has been shown to influence food intake in several murine models. Activation of GLP-1 producing cells using designer receptors activated by designer drugs (DREADDs) in an intersectional model that restricts DREADD expression to the intestinal epithelium, reduced food intake in mice (Bai et al., 2022; Hayashi et al., 2023), and a similar effect was achieved by activation of L-cells in the colon/rectum using the *Insl5* promoter to restrict DREADD expression to this distal cell population (Lewis et al., 2020). Although these studies revealed food intake suppressive effects of activating L-cells, they do not show that the effect was a result of GLP-1 itself; indeed, in the Insl5-DREADD model, food intake suppression was abolished by inhibiting PYY receptors (neuropeptide Y receptor type 2) but not GLP1R, suggesting a more important role for L-cell released PYY in the control of food intake.

The role of GIP in the control of food intake is more controversial. In mouse models, GIPR agonists reduce food intake through a pathway involving central GABA-ergic neurones (Liskiewicz et al., 2023), probably in the AP. Mirroring these results, we recently reported that chemogenetic activation of intestinal K-cells suppressed food intake in mice, and that this was abolished by GIPR antagonism, suggesting that gut-derived GIP plays a role in restricting food intake (Lewis et al., 2024). However, the opposing idea that endogenous GIP is pro-adipogenic is supported by findings that mice lacking *Gipr* are protected against diet-induced obesity (Miyawaki et al., 2002) and that humans with loss of function *GIPR* variants have a lower average body mass index (Kizilkaya et al., 2024). In

diet-induced obese mice and obese non-human primates GIPR antagonism acutely reduced food intake (Killion et al., 2018), leading to the question of how both GIPR agonism and GIPR antagonism can reduce feeding. Most probably, this reflects the complexity of signals controlling post-ingestive sensations and behaviours, which include gastric stretch, fullness and nausea, as well as intestinal signals that drive the consumption of nutritious foods (Tan et al., 2020). These signals converge at the level of the brainstem and hypothalamus, where there are a number of populations of inhibitory and excitatory GIPR neurones (Adriaenssens et al., 2023; Zhang et al., 2022), potentially enabling GIP to differentially affect orexigenic as well as anorexigenic signalling (Fig. 2).

In addition to modulating the amount eaten of a single available food, both incretins have also been implicated in food choice. GLP1R agonists are known to induce nausea in humans, which at least in part is a result of the activation of glutamatergic neurons in the AP (Adams et al., 2018). Nausea and even vomiting are side effects of many gut hormones at high concentrations, including GLP-1 and PYY. In mice, which cannot vomit, negative effects of these hormones are usually assessed as food preference or avoidance. Interestingly, GIPR agonists, rather than inducing avoidance, can ameliorate nauseating/avoidance effects of other gut hormones and even prevent vomiting in the house shrew triggered by GLP-1 or cisplatin (Borner et al., 2021; Borner et al., 2023). The importance of different gut hormones in food choice decisions is however quite complicated because, despite the nauseating effects of peripheral GLP1R agonists, chemogenetic activation of GLP-1 producing cells in the intestine increased consumption of a paired flavour compared to control animals (Bai et al., 2022). A similar intake preference was seen for flavours paired with chemogenetic activation of CCK-expressing cells in the intestine (Bai et al., 2022) and CCK-expressing cells were also linked to the development of preference for sugar or fat-containing foods over non-nutritious alternatives (Buchanan et al., 2022; Li et al., 2022; Tan et al., 2020), despite both hormones long being established to have anorexic outcomes in the absence of choice.

**Agonists of GLP1R and GIPR in the treatment of type 2 diabetes and obesity.** A wealth of preclinical and clinical data has supported and accompanied the development of GLP-1-based therapies for type 2 diabetes and obesity (Gabery et al., 2020; Secher et al., 2014), a market now worth billions of dollars per year. Although originally focussed around GLP-1, research in recent years has demonstrated improved clinical efficacy by coadministering GLP1R agonism with drugs targeting different receptors, including GIPR, GCGR and amylin receptors. This can be achieved either by coadministration of individual single-receptor agonists (e.g. a GLP1R agonist with an amylin receptor agonist) (Liberini et al., 2019) or by engineering peptides that have dual or triple agonist activity against multiple receptors, such as GLP1R/GIPR (Rosenstock et al., 2021), GLP1R/GCGR (Bluher et al., 2024; le Roux et al., 2024) and GLP1R/GIPR/GCGR (Jastreboff et al., 2023). The development of the GLP1R/GIPR dual agonist tirzepatide brought new insights into the role of GIP in the treatment of T2D and obesity because it can reduce both plasma

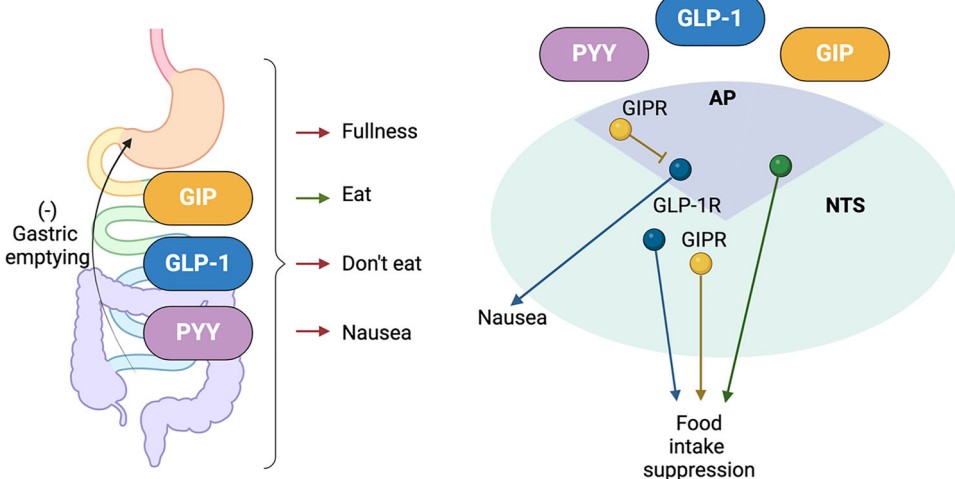

**Figure 2. Gut brain signalling**
Hormonal and neuronal signals from the gut underlie a variety of sensations related to the control of food ingestion. A number of gut hormonal signals converge at the area postrema (AP) and nucleus of the solitary tract (NTS) in the brainstem, where different neuronal populations (represented by different colours) underlie nausea and food intake suppression. GIPR positive neurones in the AP are predominantly GABA-ergic and inhibit other pathways such as GLP-1-induced nausea. Further characterisation of the neurocircuitry in the AP and NTS, as well as nuclei in the hypothalamus (not shown), underlying food intake regulation is an area of intense scientific focus.

glucose and glycated haemoglobin more effectively than the GLP1R agonist semaglutide. Importantly, tirzepatide also induces body weight loss (Gallwitz, 2022).

GIPR is expressed in white adipose tissue, predominantly in pericytes controlling blood flow, and mesothelial cells which might act as adipocyte precursors (Campbell et al., 2022). Although some studies have demonstrated GIPR antagonism by long-acting GIPR-agonists in adipocytes (Killion et al., 2020) and others have questioned the functional expression of GIPR in adipocytes themselves (Campbell et al., 2022), recent studies with tirzepatide and/or a GIPR-only agonist suggest that GIPR agonism enhances insulin signalling, glucose uptake and the conversion of glucose into glycerol, whereas, in the absence of insulin, GIPR agonism increases lipolysis. This suggests that long-term GIPR agonism can modulate adipose tissue metabolism differently in the fed and fasted states (Manchanda & Tomas, 2024; Regmi et al., 2024; Samms et al., 2021).

Reflecting the complexity of GIPR physiology described above, preclinical and clinical evidence has shown that improved efficacy on weight loss can be achieved by adding either a GIPR agonist or GIPR antagonist on top of a GLP1R targeted drug (Jensen et al., 2024). Much research and debate have gone into uncovering the explanation for these observations, with current ideas highlighting the multiple sites of action of GIP across tissues ranging from pancreatic islets and adipose tissue stores to distinct nuclei in the brainstem and hypothalamus. In the context of a desired clinical outcome of reducing blood glucose and body weight, it may be that GIPR has opposing actions that need to be balanced.

## Conclusions

Both incretin hormones, GLP-1 and GIP, play important roles in nutrient homeostasis. Their classical role as incretins helps to adjust insulin secretion in response to glucose ingestion. Both hormones, however, also have additional functions, with GLP-1 for example slowing gastric emptying, thereby reducing the speed of nutrient absorption after a meal, and GIP playing a role in fatty acid disposal into white adipose tissue. Both hormones also regulate food intake, and the success of GLP1R agonists and dual GLP1R/GIPR agonists in the treatment of obesity has kindled ongoing research to better understand which target cells are most relevant for these pharmacological outcomes.

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

## Additional information

### Competing interests

FMG and FR have received grant funding for separate projects from AstraZeneca and Eli Lilly. They received sponsorship for hosting the European Incretin Study Group meeting in Cambridge 2024 from Eli Lilly, AstraZeneca, Sun Pharma and Mercodia.

### Author contributions

C.A., F.R. and F.M.G. conceptualised, wrote and revised the manuscript. All authors have approved the final version of the manuscript submitted for publication and agree to be accountable for all aspects of the work. All persons designated as authors qualify for authorship, and all those who qualify for authorship are listed.

### Funding

This research was funded by a Wellcome joint investigator award to FR/FMG (220271/Z/20/Z) and the MRC-Metabolic Diseases Unit (MRC_MC_UU_12012/3).

## Keywords

enteroendocrine, GIP, GLP-1, gut hormone, incretin, obesity

## Supporting information

Additional supporting information can be found online in the Supporting Information section at the end of the HTML view of the article. Supporting information files available:

**Peer Review History**

