## [Peer Review History · The Journal of Physiology]

Incretin hormones and Obesity

Constanza Alcaino, Frank Reimann, and Fiona Gribble

DOI: 10.1113/JP286293

Corresponding author(s): Fiona Gribble (fmg23@cam.ac.uk)

Review Timeline:

Submission Date:	28-Aug-2024
Editorial Decision:	27-Sep-2024
Revision Received:	25-Oct-2024
Accepted:	31-Oct-2024

Senior Editor: Kim Barrett

Reviewing Editor: Stephen Keely

Transaction Report:

Dear Dr Gribble,

Re: JP-SR-2024-286293 "Incretin hormones and Obesity" by Constanza Alcaino, Frank Reimann, and Fiona Gribble

Thank you for submitting your manuscript to The Journal of Physiology. It has been assessed by a Reviewing Editor and by 2 expert referees and we are pleased to tell you that it is acceptable for publication following satisfactory revision.

ABSTRACT FIGURES: Authors may use The Journal's premium BioRender account to create/redraw their Abstract Figures (and any other suitable schematic figures). Information on how to access this account is here: <https://physoc.onlinelibrary.wiley.com/journal/14697793/biorender-access>.

REVISION CHECKLIST: Upload a full Response to Referees file. To create your 'Response to Referees' copy all the reports, including any comments from the Senior and Reviewing Editors, into a Microsoft Word, or similar, file and respond to each point, using font or background colour to distinguish comments and responses and upload as the required file type.

- 'Potential Cover Art' for consideration as the issue's cover image.
- Appropriate Supporting Information (Video, audio or data set: see https://jp.msubmit.net/cgi-bin/main.plex?form_type=display_requirements#supp).

We look forward to receiving your revised submission.

Yours sincerely,

Kim Barrett
Senior Editor

REQUIRED ITEMS

- Please include an Abstract Figure file, as well as the Figure Legend text within the main article file. The Abstract Figure is a piece of artwork designed to give readers an immediate understanding of the Review Article and should summarise the main conclusions. If possible, the image should be easily 'readable' from left to right or top to bottom. It should show the physiological relevance of the Review so readers can assess the importance and content of the article. Abstract Figures should not merely recapitulate other figures in the Review. Please try to keep the diagram as simple as possible and without superfluous information that may distract from the main conclusion of the Review. Abstract Figures must be provided by authors no later than the revised manuscript stage and should be uploaded as a separate file during online submission labelled as File Type 'Abstract Figure'. Please ensure that you include the figure legend in the main article file. All Abstract Figures will be sent to a professional illustrator for redrawing and you may be asked to approve the redrawn figure before your paper is accepted.

- Your MS must include a complete "Additional information section" with the following 4 headings and content:

Competing Interests: A statement regarding competing interests. If there are no competing interests, a statement to this effect must be included. All authors should disclose any conflict of interest in accordance with journal policy.

Author contributions: Each author should take responsibility for a particular section of the study and have contributed to writing the paper. Acquisition of funding, administrative support or the collection of data alone does not justify authorship; these contributions to the study should be listed in the Acknowledgements. Additional information such as 'X and Y have contributed equally to this work' may be added as a footnote on the title page.

It must be stated that all authors approved the final version of the manuscript and that all persons designated as authors qualify for authorship, and all those who qualify for authorship are listed.

Funding: Authors must indicate all sources of funding, including grant numbers. If authors have not received funding, this must be stated.

It is the responsibility of authors funded by RCUK to adhere to their policy regarding funding sources and underlying research material. The policy requires funding information to be included within the acknowledgement section of a paper. Guidance on how to acknowledge funding information is provided by the Research Information Network. The policy also requires all research papers, if applicable, to include a statement on how any underlying research materials, such as data, samples or models, can be accessed. However, the policy does not require that the data must be made open. If there are considered to be good or compelling reasons to protect access to the data, for example commercial confidentiality or legitimate sensitivities around data derived from potentially identifiable human participants, these should be included in the statement.

Acknowledgements: Acknowledgements should be the minimum consistent with courtesy. The wording of acknowledgements of scientific assistance or advice must have been seen and approved by the persons concerned. This section should not include details of funding.

- The reference list must be in alphabetical order, rather than numbered, to comply with our Journal format.

EDITOR COMMENTS

Reviewing Editor:

This is a well written and timely review and will make a fine contribution to the Journal's special issue on "The Role of the Gut in Obesity". I agree that the authors should consider both Reviewers' comments and, in particular, Reviewer 2's comments regarding the overall focus and scope for inclusion of schematic diagrams.

Please also see 'Required Items' above.

REFEREE COMMENTS

Referee #1:

This is a well-written review of the incretin hormones and obesity. It provides a timely, comprehensive and thorough update on what has become a very topical subject with the expanded use of receptor agonist drugs.

It is easy to read and clear in most of the detail. There are a few areas that may help by further clarification:

GLP-2, while not an incretin, needs to be introduced before p8.

Also the contribution of oxyntomodulin should be made clearer.

p10 -- MAFLD is now known as MASLD

The literature is moving very quickly and the authors may wish to review some of the most recent publications.

Referee #2:

The review "Incretin hormones and Obesity" by Alcaïno et al. is an excellent review of the recent literature with well-chosen papers, with a highly relevant graphical abstract that provides a simple didactic answer to the question of incretins and obesity.

However, the title and the graphical summary are very limited compared with the mass of bibliographical information reported in this review, which could then be considered irrelevant as it often relates to other intestinal hormones and to another physiopathological condition, diabetes. Of course, it would be out of the question not to present the whole spectrum of enteroendocrine cells and the peptides they produce, or the diabetes closely linked to obesity, but this could be seen as confusing the message. The reader might even be disappointed by the lack of schematic summaries of the different enteroendocrine cells and the different molecular mechanisms of nutrient sensing.... Similarly, molecules never before mentioned appear on page 13 or 14 (GDF15, amylin, beta-arrestin)...

The reviewer therefore suggests that, if the title and graphical design remain as initially conceived, the review should focus more quickly (from line 232) on GLP-1 and GIP and their role in the controls of food intake and body weight, with a few appropriate figures, and in particular on the potential anti-obesity mechanisms of GLP-1R and GIPR agonists.

END OF COMMENTS

Confidential Review

28-Aug-2024

The review "Incretin hormones and Obesity" by Alcaino *et al.* is an excellent review of the recent literature with well-chosen papers, with a highly relevant graphical abstract that provides a simple didactic answer to the question of incretins and obesity.

However, the title and the graphical summary are very limited compared with the mass of bibliographical information reported in this review, which could then be considered irrelevant as it often relates to other intestinal hormones and to another physiopathological condition, diabetes. Of course, it would be out of the question not to present the whole spectrum of enteroendocrine cells and the peptides they produce, or the diabetes closely linked to obesity, but this could be seen as confusing the message. An uninitiated reader might even be disappointed by the lack of schematic summaries of the different enteroendocrine cells and the different molecular mechanisms of nutrient sensing.... Similarly, molecules never before mentioned appear on page 13 or 14 (GDF15, amylin, beta-arrestin)...

The reviewer therefore suggests that, if the title and graphical design remain as initially conceived, the review should focus more quickly (from line 232) on GLP-1 and GIP and their role in the controls of food intake and body weight, with a few appropriate figures, and in particular on the potential anti-obesity mechanisms of GLP-1R and GIPR agonists.

We would like to thank the reviewers for their thoughtful comments and suggestions, which are addressed below and in the revised manuscript.

Referee #1:

This is a well-written review of the incretin hormones and obesity. It provides a timely, comprehensive and thorough update on what has become a very topical subject with the expanded use of receptor agonist drugs.

It is easy to read and clear in most of the detail. There are a few areas that may help by further clarification:

GLP-2, while not an incretin, needs to be introduced before p8.

Response: We have moved this section to page 5, when introducing EEC hormones.

Also the contribution of oxyntomodulin should be made clearer.

Response: We have explained that its effects on metabolism are due to its actions as a dual agonist of GLP1R and GCCR.

p10 -- MAFLD is now known as MASLD

Response: This has been corrected.

The literature is moving very quickly and the authors may wish to review some of the most recent publications.

Response: We have incorporated a section on GIP action on adipose tissue from the most recent literature (Regmi et al., 2024; Manchanda & Tomas 2024), and have included more detail of the action of GIP on food intake. The new sections of text read as follows:

“GIPR is expressed in white adipose tissue, predominantly in pericytes controlling blood flow, and mesothelial cells which might act as adipocyte precursors (Campbell et al., 2022). Whilst some have demonstrated antagonism by long acting GIPR-agonists in adipocytes (Killion et al., 2020) and others have questioned functional expression of GIPR in adipocytes themselves (Campbell et al., 2022), recent studies with tirzepatide and/or a GIPR-only agonist suggest that GIPR agonism enhances insulin signalling, glucose uptake and the conversion of glucose into glycerol, whilst in the absence of insulin GIPR agonism increases lipolysis. This suggests that long-term GIPR agonism can modulate adipose tissue metabolism differently in the fed and fasted states (Samms et al., 2021; Manchanda & Tomas, 2024; Regmi et al., 2024).”

And:

“The role of GIP in the control of food intake is more controversial. In mouse models, GIPR agonists reduce food intake through a pathway involving central GABA-ergic neurones (Liskiewicz et al., 2023), likely in the area postrema. Mirroring these results, we recently reported that chemogenetic activation of intestinal K-cells suppressed food intake in mice, and that this was abolished by GIPR antagonism, suggesting that gut-derived GIP plays a role in restricting food intake (Lewis et al., 2024). However, the opposing idea that endogenous GIP is pro-adipogenic is supported by findings that mice lacking GIPR are protected against diet induced obesity (Miyawaki et al., 2002) and that humans with loss of function GIPR variants have a lower average body mass index (Kizilkaya et al., 2024). In diet-induced obese mice and obese non-human primates GIPR antagonism acutely reduced food intake (Killion et al., 2018), leading to the question of how both GIPR agonism and GIPR antagonism can reduce feeding. Likely this reflects the complexity of signals controlling post-ingestive sensations and behaviours, which include gastric stretch, fullness and nausea, as well as intestinal signals that drive consumption of nutritious foods (Tan et al., 2020). These signals converge at the level of the brainstem and hypothalamus, where there are a number of populations of inhibitory and excitatory GIPR neurones (Zhang et al.,

2022; Adriaenssens et al., 2023), potentially enabling GIP to differentially affect orexigenic as well as anorexigenic signalling (Figure 2).”

Referee #2:

The review "Incretin hormones and Obesity" by Alcaino et al. is an excellent review of the recent literature with well-chosen papers, with a highly relevant graphical abstract that provides a simple didactic answer to the question of incretins and obesity.

However, the title and the graphical summary are very limited compared with the mass of bibliographical information reported in this review, which could then be considered irrelevant as it often relates to other intestinal hormones and to another physiopathological condition, diabetes. Of course, it would be out of the question not to present the whole spectrum of enteroendocrine cells and the peptides they produce, or the diabetes closely linked to obesity, but this could be seen as confusing the message. The reader might even be disappointed by the lack of schematic summaries of the different enteroendocrine cells and the different molecular mechanisms of nutrient sensing.... Similarly, molecules never before mentioned appear on page 13 or 14 (GDF15, amylin, beta-arrestin)...

Response: We appreciate the reviewer's thoughts on schematics and have added a new Figure 1 (Nutrient sensing by EECs) and Figure 2 (GIP and GLP-1 in the CNS). We have removed mention of beta-arrestin and GDF15, as they are not directly relevant to the focus of the review and are not mentioned elsewhere. However, we have retained the references to amylin, which although not an incretin itself, is a hot topic in the GLP-1 based drug development field.

The reviewer therefore suggests that, if the title and graphical design remain as initially conceived, the review should focus more quickly (from line 232) on GLP-1 and GIP and their role in the controls of food intake and body weight, with a few appropriate figures, and in particular on the potential anti-obesity mechanisms of GLP-1R and GIPR agonists.

Response: We have retained the title and graphical abstract, as originally submitted, but have added two additional figures to enhance the text. We have also modified the text after line 232 to address more clearly and separately the roles of GIPR and GLP1R in food intake regulation and obesity.

Dear Professor Gribble,

Re: JP-SR-2024-286293R1 "Incretin hormones and Obesity" by Constanza Alcaino
Frank Reimann
Fiona Gribble

I am pleased to tell you that your Symposium Review article has been accepted for publication in The Journal of Physiology, subject to any modifications to the text that may be required by the Journal Office to conform to House rules.

NEW POLICY: In order to improve the transparency of its peer review process, The Journal of Physiology publishes online as supporting information the peer review history of all articles accepted for publication. Readers will have access to decision letters, including all Editors' comments and referee reports, for each version of the manuscript and any author responses to peer review comments. Referees can decide whether or not they wish to be named on the peer review history document.

The last Word version of the paper submitted will be used by the Production Editors to prepare your proof. When this is ready, you will receive an email containing a link to Wiley's Online Proofing System. The proof should be checked and corrected as quickly as possible.

All queries at proof stage should be sent to tjp@wiley.com.

The accepted version of the manuscript is the version that will be published online until the copy edited and typeset version is available. Authors should note that it is too late at this point to offer corrections prior to proofing. Major corrections at proof stage, such as changes to figures, will be referred to the Reviewing Editor for approval before they can be incorporated. Only minor changes, such as to style and consistency, should be made a proof stage. Changes that need to be made after proof stage will usually require a formal correction notice.

Are you on Twitter? Once your paper is online, why not share your achievement with your followers. Please tag The Journal (@jphysiol) in any tweets and we will share your accepted paper with our 30,000+ followers!

If you would like to receive our 'Research Roundup', a monthly newsletter highlighting the cutting-edge research published in The Physiological Society's family of journals (The Journal of Physiology, Experimental Physiology and Physiological Reports), please click this link, fill in your name and email address and select 'Research Roundup':
<https://www.physoc.org/journals-and-media/membernews/>

Yours sincerely,

Kim Barrett
Senior Editor
The Journal of Physiology

EDITOR COMMENTS

Reviewing Editor:

Thanks to the authors for fully taking the reviewers' comments on board. The new Figures are a particularly nice addition. I'm sure this review will be of real interest to the readership of the Journal of Physiology and will make an excellent contribution to the upcoming special issue.

Senior Editor:

Thank you for this exceptional contribution to our special issue.

*** IMPORTANT NOTICE ABOUT OPEN ACCESS ***

To assist authors whose funding agencies mandate public access to published research findings sooner than 12 months after publication, The Journal of Physiology allows authors to pay an open access (OA) fee to have their papers made freely available immediately on publication.

You will receive an email from Wiley with details on how to register or log-in to Wiley Authors Services where you will be able to place an OnlineOpen order.

You can check if your funder or institution has a Wiley Open Access Account here: <https://authorservices.wiley.com/author-resources/Journal-Authors/licensing-and-open-access/open-access/author-compliance-tool.html>.

Your article will be made Open Access upon publication, or as soon as payment is received.

If you wish to put your paper on an OA website such as PMC or UKPMC or your institutional repository within 12 months of publication you must pay the open access fee, which covers the cost of publication.

OnlineOpen articles are deposited in PubMed Central (PMC) and PMC mirror sites. Authors of OnlineOpen articles are permitted to post the final, published PDF of their article on a website, institutional repository, or other free public server, immediately on publication.

Note to NIH-funded authors: The Journal of Physiology is published on PMC 12 months after publication, NIH-funded authors DO NOT NEED to pay to publish and DO NOT NEED to post their accepted papers on PMC.

1st Confidential Review

25-Oct-2024